# Host-Strain-Specific Responses to Pneumonia Virus of Mice Infection: A Study of Lesions, Viral Load, and Cytokine Expression

**DOI:** 10.3390/v17040548

**Published:** 2025-04-09

**Authors:** Etienne Levy, Gautier Gilliaux, Michaël Sarlet, Daniel Desmecht, Anne-Sophie Van Laere

**Affiliations:** Department of Pathology, FARAH Research Center, Faculty of Veterinary Medicine, University of Liège, 4000 Liège, Belgium; elevy@uliege.be (E.L.); gautiergilliaux@hotmail.com (G.G.); michael.sarlet@uliege.be (M.S.); daniel.desmecht@uliege.be (D.D.)

**Keywords:** pneumonia virus of mice, respiratory syncytial virus, cytokine, mouse line

## Abstract

Pneumonia virus of mice (PVM) infection is a reference animal model for human respiratory syncytial virus (hRSV), a leading cause of lower respiratory tract disease in children under 5 years of age and in the elderly. This longitudinal study employed necropsy to examine macroscopic lesions, histological slides to assess microscopic lesions, and qRT-PCR to measure lung viral load and cytokine expression in PVM-infected mice from three different genetic backgrounds, spanning from day 1 to day 6 post-infection. Our analysis reveals a strong correlation between viral load and microscopic lesions across the 129/Sv, BALB/c, and SJL/J mouse lines, indicating that PVM pathogenicity is partially driven by the virus itself. Additionally, a significant correlation between cytokine levels and lesion severity was observed in 129/Sv and BALB/c mice, suggesting an important role of cytokines in disease progression. This study emphasizes the interplay between viral load and cytokine-driven tissue damage, with genetic background significantly influencing disease outcomes.

## 1. Introduction

Human respiratory syncytial virus (hRSV) is one of the main causes of lower respiratory tract disease in children younger than 5 years of age and elderly people. This ubiquitous pathogen infects nearly all children before their second birthday [1] and is the most common cause of hospitalization for infants [2]. The pooled incidence of mild (cough, nasal congestion, rhinorrhea, fever, vomiting, and diarrhea), moderate (shortness of breath, dyspnea, wheezing, pneumonia, need for oxygen supplementation), and severe (respiratory failure, sepsis, need to be admitted to the intensive care unit) clinical manifestations were, respectively, quantified as 51, 37, and 7% [3]. A recent review [4] estimated that globally, in 2019, there were 33 million RSV-associated acute lower respiratory infection episodes, leading to 3.6 million RSV-associated acute lower respiratory infection hospital admissions and 26,300 RSV-associated acute lower respiratory infection in-hospital deaths. Furthermore, they concluded that, in 2019, more than 100,000 deaths in children aged 0–5 years were attributable to RSV.

Pneumonia virus of mice (PVM) infection in mice is the reference animal model for hRSV, as it has been shown to faithfully reproduce many of the clinical and pathologic features of the more severe forms of RSV infection in infants [5]. PVM is a natural rodent pathogen. Like hRSV, PVM is an enveloped, negative-sense, single-stranded RNA virus of the family *Paramyxoviridae*, subfamily *Pneumovirinae*, genus *Pneumovirus*. Its genome is 14,886 nucleotides long and codes for 10 mRNAs translated into 12 proteins [6]. PVM is tropic for the respiratory epithelium and transmitted exclusively horizontally via the respiratory tract, mainly by direct contact and aerosol [7].

Genetic background has often been mentioned among the known factors influencing the clinical severity of RSV [8,9,10]. Similarly, it has been shown that susceptibility to viral diseases [11], including PVM [12,13], differs dramatically among inbred mouse strains. A pattern of continuous variation in resistance/susceptibility has been described in mice with, among others, resistant SJL/J, intermediate BALB/c, and susceptible 129/Sv strains. Inbred mice lines are thus an interesting model for dissecting mechanisms that lead to different disease patterns observed in humans.

In this longitudinal study, we characterized these differences in terms of clinical outcome and macroscopic and microscopic lung lesions during the course of the disease. In addition, we characterized the expression profile of 15 cytokines in the lungs of infected mice 1–4 and 6 days post-infection (d.p.i.). We aimed to identify cytokines with different expression patterns among the studied susceptible, intermediate, and resistant mouse lines, as well as to assess whether and to what extent the cytokine storm phenomenon contributed to the pathogenicity of pneumoviruses.

## 2. Material and Methods

### 2.1. Mice

Fifty 7-week-old female mice of each of the 3 studied inbred lines (129S2/SvPasOrlRj, BALB/cJRj, and SJL/JRj) were purchased from Janvier Labs (Le Genest-Saint-Isle, France) and acclimated for one week in a BSL2 laboratory before the beginning of the experiment. Mice were housed in a manner that complied with the National Institute of Health guidelines and the official Belgian guidelines. Mice were randomly assigned into groups of 5 animals upon arrival and were kept in disposable cages (Innovive, France). Enrichment included nesting material (Carfil Cotton, Carfil Quality, Belgium), shelter (Carfil Dome Mini, Carfil Quality, Belgium), and Carfil ES-Brick (Carfil Quality, Belgium). The controlled laboratory conditions were temperature (22 ± 2 °C), humidity (50 ± 10%), and a 12 h light–dark cycle, with ad libitum access to a commercial diet (Rats & Mice Maintenance, Carfil Quality, Belgium) and tap water. One group of five PVM-infected mice of each inbred strain was euthanized on days 1 to 6 post-infection. The group size was determined using the G*Power 3.1.9.7 software. A group size of five individuals allowed for the detection of a statistically significant difference in cytokines whose expression was at least doubled, with α = 0.05 and a power of 0.85. One SJL/J mouse was excluded during the study because its viral load was more than seventy times lower than that of the other four mice in its group. This indicated that the mouse had not been properly infected. Consequently, the SJL/J group euthanized at 3 d.p.i. consisted of only four animals. This study was approved by the University of Liege Ethics Committee under reference 21-2311.

### 2.2. Viral Infection

A highly virulent stock of PVM strain J3666 was prepared by homogenizing mouse lungs with a blade after serial passages in 8-week-old mice. The stock was propagated in DMEM supplemented with 10% FBS, then clarified by centrifugation, filtered, dialyzed, aliquoted, and stored in liquid nitrogen until use. Infectious titers were measured using the standard median tissue culture infectious dose (TCID_50_) assay on BHK-21 cells. Titration of randomly selected aliquots yielded highly reproducible results. A total of 50 µL of PBS containing 10 TCID_50_ of the pathogenic strain of Pneumovirus of Mice (J3666) was intranasally instilled under gaseous anesthesia (Isoflurane).

### 2.3. Clinical Score

Following the infection, we recorded daily longitudinal clinical data, including appearance, behavior, body condition, and temperature. Temperature was measured with a Braun™ NTF 3000 infrared thermometer, aiming for the perineal area, as described before [14]. Mice were weighed on a BL310 scale (Sartorius) in a 10 × 8 × 10 cm transparent plastic box. The weights were recorded once the individuals were still. The clinical scores corresponding to each of the evaluated parameters are given in Table 1. Mice were euthanized if they reached a total score ≥ 6 or if they reached one of the following end-points: (1) live weight loss greater than or equal to 20%; (2) adoption of a discordant expiratory pattern; or (3) signs of suffering detected by facial appearance assessment, hair condition, ear and mustache position, social behavior change, and decreased movement in the cage.

### 2.4. Macroscopic Lesions

Mice were euthanized by ketamine and xylazine overdosing. Lungs were removed from the thoracic cavity after confirmation of death, quickly dripped into a sterile Ringer’s lactate solution, and dried with a soft cloth. A photographic setup was used, including a tripod maintaining the camera above the lungs. The camera used was a Pentax K-1 mark II with a 100 mm f/2.8 macro lens. White balance was set using an X-Rite White Balance calibrated card. The light source was exclusively artificial with an unknown color rendering index. The settings were as follows: ISO200–F/5.6-1/12th second—distance to the subject: 60 cm. RAW pictures were developed with DXO Photolabs 2 to crop each shot in a TIFF file. All the TIFF files were assembled using Sherif Affinity Photos 2 to produce the final PNG picture.

### 2.5. Microscopic Lesions

Left lungs were fixed with formaldehyde for 24 h, then stored in 70% ethanol. Inclusion was performed in a Leica™ Histocore Pearl, embedding was performed in a Leica™ Histocore Arcadia H + C. Four µm slices were cut with a Leica™ RM2235 microtome and left to dry overnight on a Leica™ HI1220 flattening table. Hematoxylin and eosin (H&E) staining protocol was programmed on a Myr™ Myreva SS-30. The slide and the coverslide were glued using a Merck™ neo-mount resin. Pictures were recorded with an Olympus DP-73 camera in high-resolution binning mode on Cellsens entry software, generating a TIFF file. Sharpness was enhanced to 100 on Dxo Photo Labs 0, and then an unsharpen mask was applied on Sherif Affinity Photos 2 with the following settings: radius 64.40, Factor 1.009, and Threshold 1%. This was carried out to compensate for the low resolution of the original picture on high magnification. All the TIFF files were assembled using Sherif Affinity Photos 2 to produce the final PNG picture. Microscopic lesions were scored based on the percentage of affected fields. Each slide was meticulously examined at 10× magnification, field by field, with higher magnification used when necessary to identify and confirm lesions. The scoring scale ranged from 0, indicating lesions in 0–5% of fields, to 3, indicating lesions in more than 95% of fields. Intermediate scores were assigned as follows: 0.5 for 5–25%, 1 for 25–50%, 1.5 for 50%, 2 for 50–75%, and 2.5 for 75–95% of fields. To ensure an unbiased evaluation, the examiner was blinded to the mouse line, infection status, and time of euthanasia.

### 2.6. qRT-PCR

Right lungs were flash-frozen in liquid nitrogen and stored at −80 °C. Total RNA was extracted with TRIzol reagent (Invitrogen, Carlsbad, CA, USA.) according to the manufacturer’s protocol. Contaminating DNA was removed with the Turbo DNA-free kit (Invitrogen). For cytokine quantification, qRT-PCR was performed in duplicate or triplicate as follows: 1 × Luna Universal Probe One-Step Reaction Mix, 1 × PrimeTime qPCR Assay (IDT), 0.75 µL Luna WarmStart RT Enzyme Mix, and 2 µL of RNA in a total of 15 µL. Primer and probe sequences are given in Table 2. Cycling was performed on an AbiPrism 7900 HT (Applied Biosystems, Foster City, CA, USA) with the following program: 55 °C for 10 min; 95 °C for 1 min; 40 cycles of 15 s at 95 °C, 1 min at 60 °C. Relative gene expression was normalized to GAPDH and RPLPo expression. These two reference genes were chosen because they had the lowest M value in a preliminary geNorm analysis initiated on 15 samples (one mouse from each of the 3 lines at each of the 5 times points) and 5 reference targets. qRT-PCR results were analyzed with qBase+ version 3.4 (CellCarta, Montreal, QC, Canada). The two same samples were added to each plate to allow interrun calibration. qRT-PCR results were analyzed in qBase+ v3.4 (CellCarta). The statistical significance of the difference between groups was assessed by one-way ANOVA with post hoc Tukey HSD. We were unable to analyze IL-4 expression from one SJL/J mouse at day 3 p.i. due to insufficient remaining RNA. Viral particle quantification was performed by amplifying a segment of the gene encoding the surface hydrophobic (SH) protein using the primers PVM.SH.F (GCCGTCATCAACACAGTGTGT) and PVM.SH.R (GCCTGATGTAGCAATGCTT), along with the probe PVM.probe (/56-FAM/CGCTGATAA/ZEN/TGGCCTGCAGCA/3IABkFQ). PCR reactions were conducted in triplicate in a 20 µL volume containing 1× Luna Universal Probe One-Step Reaction Mix, 1 µL Luna WarmStart RT Enzyme Mix, 0.8 µM of each primer, 0.4 µM probe, and 50 ng of RNA. Amplification was performed on a StepOnePlus (Applied Biosystems) under the following conditions: 10 min at 55 °C, 1 min at 95 °C, followed by 40 cycles of 10 s at 95 °C and 1 min at 60 °C. To enable absolute quantification, serial dilutions of a synthetic RNA corresponding to the target region were amplified in parallel to establish a standard curve. The statistical significance of the difference between groups was assessed by one-way ANOVA with post hoc Tukey HSD.

### 2.7. Correlation Analyses, Principal Component Analyses (PCA), and Heatmaps

Correlation analyses (Pearson’s r) were performed in Past 4.03. Bonferroni corrections were applied to account for multiple tests.

PCA analyses were performed with ClustVis (https://biit.cs.ut.ee/clustvis/, accessed on 26 July 2024). The following parameters were applied: original values were ln (x + 1)-transformed, no scaling was applied to rows, and SVD with imputation was used to calculate principal components.

Heatmaps were generated with ClustVis (https://biit.cs.ut.ee/clustvis/, accessed on 26 July 2024). Applied parameters are as follows: original values are ln (x + 1)-transformed; rows are centered; no scaling is applied to rows; imputation is used for missing value estimation; and both rows and columns are clustered using correlation distance and average linkage.

## 3. Results

The genetic background of mice has a significant impact on clinical outcomes and lesions caused by a PVM infection.

To describe the evolution of the disease in three commercially available mouse lines, we inoculated 129Sv, BALB/c, and SJL/J mice with 10 TCID_50_ of the pathogenic strain of PVM virus (J3666). Following the infection, we recorded longitudinal clinical data, including body weight, skin temperature, respiratory pattern, fur condition, and posture. Mice were euthanized if they reached one of the end-point thresholds or if their cumulative clinical score was equal to or higher than 6. All mice had a score of 0 on days 1–4. Differences between lines appeared on day 5, when the clinical condition of 129/Sv and BALB/c but not of SJL/J mice started to degrade. These differences were enhanced on day 6 p.i., with 14 129/Sv and 4 BALB/c but no SJL/J mice reaching a cumulative clinical score exceeding 6, i.e., reaching the threshold for euthanasia (Figure 1). Furthermore, at this time point, the three mouse strains differ significantly from each other, as determined by a one-way ANOVA followed by a post hoc Tukey HSD test (*p* < 0.01 for each of the three pairwise comparisons).

### 3.1. The Genetic Background of Mice Has a Significant Impact on Macroscopic Lesions

We observed diffuse lung congestion from day 1 to day 3 p.i. in all mice (Figure 2). The clinical status of 129/Sv and BALB/c mice evolved badly starting from day 3 p.i. with diffuse multilobular pneumonia, worsening to a severe diffuse pneumonia. These lesions are typical of a severe PVM infection [15]. Meanwhile, in SJL/J mice, the lung congestion decreases between day 3 and 6 p.i. Lungs of mock-infected animals show a faint congestion, compatible with the euthanasia procedure.

### 3.2. The Genetic Background of Mice Has a Significant Impact on Microscopic Lesions

Macroscopic observations were confirmed by microscopic findings (Figure 3, Table 3). The lungs of infected mice are congested as a consequence of the diffuse interstitial inflammation. This inflammation is less severe in SJL/J mice compared with 129/Sv and BALB/c mice at all time points. It is first associated with a neutrophilic infiltration, followed by a lymphocytic infiltration.

A perivascular inflammation leading to perivascular edema can be observed in 129/Sv and BALB/c mice from day 5 p.i. Inflammation spreads through the parenchyma in the most susceptible mice, causing necrosis and microhemorrhages. Contrariwise, the lungs of the more resistant SJL/J mice appear almost normal on days 5 and 6 p.i., displaying only a slight remaining congestion.

No significative lesion except congestion can be observed in the lungs of mock-infected mice. The latter is most likely the result of euthanasia [16].

### 3.3. The Genetic Background of Mice Has a Significant Impact on PVM Virus Replication

A fragment of the gene coding for the surface hydrophobic protein (SH) of PVM was quantified by qRT-PCR to confirm infection of the mice following intranasal inoculation of PVM and to record the kinetics of viral replication in their lungs. We thus confirmed that all mice except one had been correctly infected with the virus. The latter was excluded from this study, as explained in the Materials and Methods section. In addition, we observed a daily increase in viral load in all mouse lines. The highest daily increase was seen between 1 and 2 d.p.i. with, respectively, 130, 81, and 25 times more virus in 129/Sv, BALB/c, and SJL/J mice. Strikingly, significant differences between lines were already observed 1 d.p.i. At this time point, the lungs of BALB/c mice contain significantly more viral particles than those of 129/Sv and SJL/J mice (14,850 ± 4403; 3299 ±1573; and 7552 ± 2413, respectively). This difference persists at 2 d.p.i., with viral loads of 1,197,665 ± 275,091 in BALB/c, 428,088 ± 93,922 in 129/Sv, and 190,067 ± 51,039 in SJL/J. After this time, viral load becomes significantly higher in both 129/Sv and BALB/c mice compared with SJL/J. From 3 to 6 d.p.i., no significant differences are observed between 129/Sv and BALB/c mice (Figure 4). Interestingly, we observed a strong correlation between the pulmonary viral load and the microscopic lesion score (Pearson’s r = 0.84, *p*= 9.58 × 10^−21^, *n* = 75).

### 3.4. The Genetic Background of Mice Has a Significant Impact on Cytokine Expression

We characterized the expression of 15 cytokines in the lungs of mock- or PVM-infected mice (1–4, and 6 d.p.i.) (Figure 5, Appendix A). Compared with the basic cytokine expression level determined in mock-infected mice, we observed a drastic increase in the expression of CCL3, CXCL1, CXCL2, CXCL10, IFNβ, IFNγ, IL-1β, IL-6, IL13, TNFα starting from day 4 p.i. and in IL-10 and IL-12a from day 6 p.i. This increased expression pattern is cytokine- as well as line-specific.

At 4 d.p.i., the relative expression level of CCL3, CXCL1, CXCL2, CXCL10, IFNλ3, IL-1β, IL-6, IL-12a and TNFα is significantly higher in 129Sv and/or BALB/c mice than in SJL/J mice. This continues to be true at 6 d.p.i. for CXCL1, CXCL2, IL-6 and IL12a. On the contrary, at day 6 p.i., the expression level of IFNγ, IL-10, and IL-13 is significantly higher in SJL/J mice than in 129Sv mice and the expression level of IFNγ, IFNλ3, and IL-1β is significantly higher in SJL/J mice compared with BALB/c mice. In addition, we observed significant differences between 129Sv and BALB/c mice 6 d.p.i., with a higher relative expression of IFNλ3, IL-6, and TNFα in 129Sv mice and a higher relative expression of IL-10 and IL-13 in BALB/c mice. Although IL-4 expression level seems unexpectedly stable during the course of the disease, it is interesting to note that its expression level is systematically lower in SJL/J mice compared with 129Sv and/or BALB/c mice from day 1 to 4 p.i. A similar observation can be made for CCL5, except on day 6 p.i., when its expression level increases in the SJL/J strain and becomes significantly higher than in the 129Sv strain.

We conducted principal component analyses (PCAs) to summarize the information collected with our qRT-PCRs. Depending on the day p.i., between 60.7 and 91.5% of the variance found in our dataset is explained by PC1 and PC2 (Table 4, Figure 6 and Appendix A). For example, on day 6 after infection, PC1 accounts for 60% and PC2 for 20.2% of the variance. The cytokines with the largest loadings are given in Table 4. On day 6, for PCA1, these were IL-10 (0.54), IFNγ (0.42), and CXCL2 (−0.41), and for PCA2, IFNβ (−0.94) and CXCL2 (−0.21).

The scatterplot revealed that different mouse lines correspond to different clusters (Figure 6 and Appendix A). From 2 d.p.i., there is no overlap between the SJL-specific cluster and the two other clusters. On day 6 p.i., all three clusters are non-overlapping, and the BALB/C-cluster is located between the 129/Sv and the SJL/J clusters, as intuitively expected from the clinical burden (Figure 6a and Appendix A). Heatmap analyses at day 6 p.i. show a tight clustering of CXCL2, IL-12a, CXCL1, and IL-6 expression on one hand and of IL-10 and IFNγ expression on the other hand (Figure 6b).

Next, we conducted pooled correlation analyses (Pearson’s r) between each studied cytokine, the pulmonary viral load, and the microscopic lesion score (Table 5). We noted very strong correlations (r > 0.9) between IL-10 and IFNγ and between IL-6 and CXCL1. Interestingly, the microscopic lesion score is primarily correlated with viral load (r = 0.839) but also shows a strong correlation with CXCL1, CXCL10, IL-6, and TNFα (r > 0.7). We repeated these analyses within each strain to gain a better insight into the different immune reactions that might take place in the different strains (Appendix A). The highest correlation was again found between the microscopic lesion score and the viral load for the three mouse lines with r values of 0.86, 0.81, and 0.70 for 129/Sv, BALB/c, and SJL/J, respectively. In addition, we found an almost as high correlation between the microscopic lesion and several cytokines in 129/Sv (CCL3, r = 0.83; TNF, r = 0.83; CXCL10, r = 0.82, CXCL1, r = 0.79; IL-6, r = 0.79) and BALB/c mice (CXCL1, r = 0.80; CCL3, r = 0.79; IL-1β, r = 0.79). The viral load strongly correlates with multiple cytokines in each of the studied mouse lines. For example, TNF (r = 0.93) and IFNγ (r = 0.91) in BALB/c mice, as well as CXCL1 (r = 0.94), CXCL2 (r = 0.94), and TNF (r = 0.91) in SJL/J mice, exhibit strong correlations.

## 4. Discussion

In this study, we evaluated the macroscopic and microscopic lesions, the viral load, and the expression level of 15 cytokines in the lungs of PVM-infected mice from three commonly used inbred lines to better understand patterns of inflammation resulting from this infection in different genetic backgrounds.

Cytokines are a broad category of small polypeptides that act as intercellular mediators. They are essential for the correct functioning of the immune system and are involved in a multiplicity of pathophysiological processes. Hyperinflammatory states secondary to the excessive production of cytokines by a deregulated immune system are known as cytokine storms. The phenomenon has attracted a lot of attention since it emerged as a key aspect of the novel Coronavirus disease 2019 (COVID-19) [17]. Excessive local release of cytokines is considered to be the determinant of pathological alterations and the clinical manifestation of the acute respiratory distress syndrome observed in the later stages of COVID-19 [18]. Cytokine storms have also been shown to determine the clinical severity and consequently the lethality of other respiratory diseases, i.e., SARS, MERS, H1N1 influenza A, Spanish flu [19], and hRSV [20]. For this reason, one of the purposes of this study was to investigate the potential role of the cytokine storm during the clinical course of PVM. Our correlation analyses demonstrate the existence of a strong link between the microscopic lesions and the viral load in the three considered mouse lines. This leads us to the conclusion that, in these inbred mouse lines, the pathogenicity of PVM is at least partially caused by the virus itself. Besides, we found an almost as strong correlation between the microscopic lesions and a series of cytokines in 129/Sv and BALB/c but not in SJL/J mice, pointing toward an equally important role of cytokines in the pathogenicity of PVM in these two first lines. Among the cytokines well correlated with the microscopic lesions, we can cite CCL3, CXCL1, CXCL10, TNF, and IFN-γ, which have all been previously reported to play a role in cytokine storms [21,22].

Interferons are essential mediators of the early immune response to viral infections. Among them, Type I IFNs (IFN-α and IFN-β) serve as key components of the innate antiviral defense, constituting the first wave of immune protection against viral invasion. IFN-β transcription can be activated ∼10,000-fold in just a few hours upon activation of innate immune sensors by invading virus particles [23,24,25]. It is surprising that in all three mouse lines examined in this study, IFN-β expression begins to rise only from day 4 p.i. This suggests a delayed or blunted interferon response, particularly in 129/Sv mice, where the viral load is as high as in BALB/c mice, yet IFN-β peaks inconsistently—emerging at day 4 in only a portion of the analyzed mice.

We observed an extremely strong positive correlation between IFNγ and IL-10 expression levels in all three lines. This may seem counterintuitive at first, as IFNγ is a pro-inflammatory cytokine known to be essential for innate and adaptive immunity against viral infections [26] and crucial for CCL3-mediated neutrophil recruitment during PVM infection [27], while IL-10 is a potent immunosuppressive and anti-inflammatory cytokine [28]. The role of IL-10 is, however, much more complex than originally thought and IL-10 is now known as a pleiotropic cytokine. Hence, IFNγ has been shown to alter the anti-inflammatory activity of IL-10 by redirecting IL-10 signaling in macrophages from the Stat3 to the Stat1 signaling pathway (i.e., from anti-inflammatory to pro-inflammatory). Hence, IFNγ and IL-10 could work in synergy to increase the pro-inflammatory response against PVM infection.

The complex interplay between IFNγ and inflammation is further highlighted by studies on PVM-infected IFNγ gene-deleted BALB/c mice. Although these mice showed no differences in viral replication or survival compared with wild-type mice, they exhibited more severe lung inflammation on day 10 post-inoculation, possibly due to the overproduction of chemoattractant and eosinophil-active cytokines, including CXCL1 and CCL3 [29]. In contrast, we observed a strong positive correlation between IFNγ expression and viral load (r = 0.91), as well as between IFNγ and CXCL1 (r = 0.8) and between IFNγ and CCL3 (r = 0.95) in our wild-type BALB/c mice. These differences could be attributed to variations between BALB/c mice, as they were obtained from different providers, discrepancies in the preparation of the viral inoculum, or differences in the timing of observations (10 days p.i. vs. 1–6 days p.i.).

We also observed a remarkably strong positive correlation between expression levels of CXCL1 and IL-6, an important proinflammatory cytokine playing an adverse role in acute respiratory infection with PVM through neutrophil recruitment and fluid accumulation in lung tissue [30]. IL-6 is known as an upstream regulator of CXCL1 in inflammatory processes, linking cytokine signaling to chemokine-mediated immune cell recruitment [31,32]. Interestingly, the reverse is also true, as CXCL1 has been shown to enhance IL-6 production in a dose- and time-dependent manner [33,34]. Based on our findings, we hypothesize that during PVM infection, CXCL1 contributes to IL-6 upregulation, as its increase from the baseline observed in mock-infected mice precedes the rise in IL-6 levels. Remarkably, these two cytokines differ significantly between 129/Sv and BALB/c mice six days after PVM infection and likely contribute to the distinct, non-overlapping clustering of these mouse strains in the cytokine scatter plot (Figure 6). However, despite their differing cytokine responses, neither strain appears capable of preventing viral amplification, nor the progression of macroscopic and microscopic lesions, which appear remarkably similar in both at six d.p.i.

From day 1 p.i., we noted that the viral load in SJL/J mice is rising at a significantly slower rate than in 129/Sv and BALB/c mice. We were unable to account for this observation using any of the parameters examined in this study. Since PVM replication occurs in the bronchial epithelium, type 1 or 2 pneumocytes, and alveolar macrophages [13], this raises the question of how genetic differences between mouse lines influence viral replication dynamics in these target cells. It would be of interest to conduct additional experiments to determine where in the replication cycle of the virus the divergence arises. Due to this reduced viral amplification, the viral load in SJL/J mice at 6 d.p.i. matches that of 129/Sv and BALB/c mice at 3 d.p.i. However, the immune response triggered by this amount of viral particles is considerably greater in SJL/J at 6 d.p.i. compared with the 129/Sv and BALB/c mice at 3 d.p.i. The cytokine expression pattern observed in SJL/J mice at 6 d.p.i. corresponds to the recognized Th1-biased immune response seen in SJL/J mice, marked by heightened levels of TNF-α and IFNγ production, resulting in enhanced expression of CXCL10 and CCL3.

In conclusion, we herein present an additional example of a genetic background-dependent disease course, where the severity of injury depends, in varying proportions, on viral load and cytokine-related damage.

## Figures and Tables

**Figure 1 viruses-17-00548-f001:**
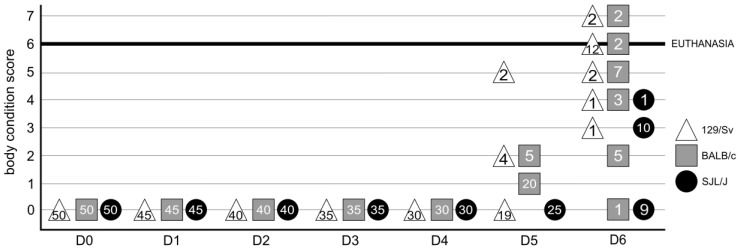
Clinical score of PVM-infected mice. The clinical condition of mice was assessed daily. The numbers written inside the triangles, squares, or circles, respectively, indicate the number of 129/Sv, BALB/c, and SJL/J mice with the corresponding clinical score.

**Figure 2 viruses-17-00548-f002:**
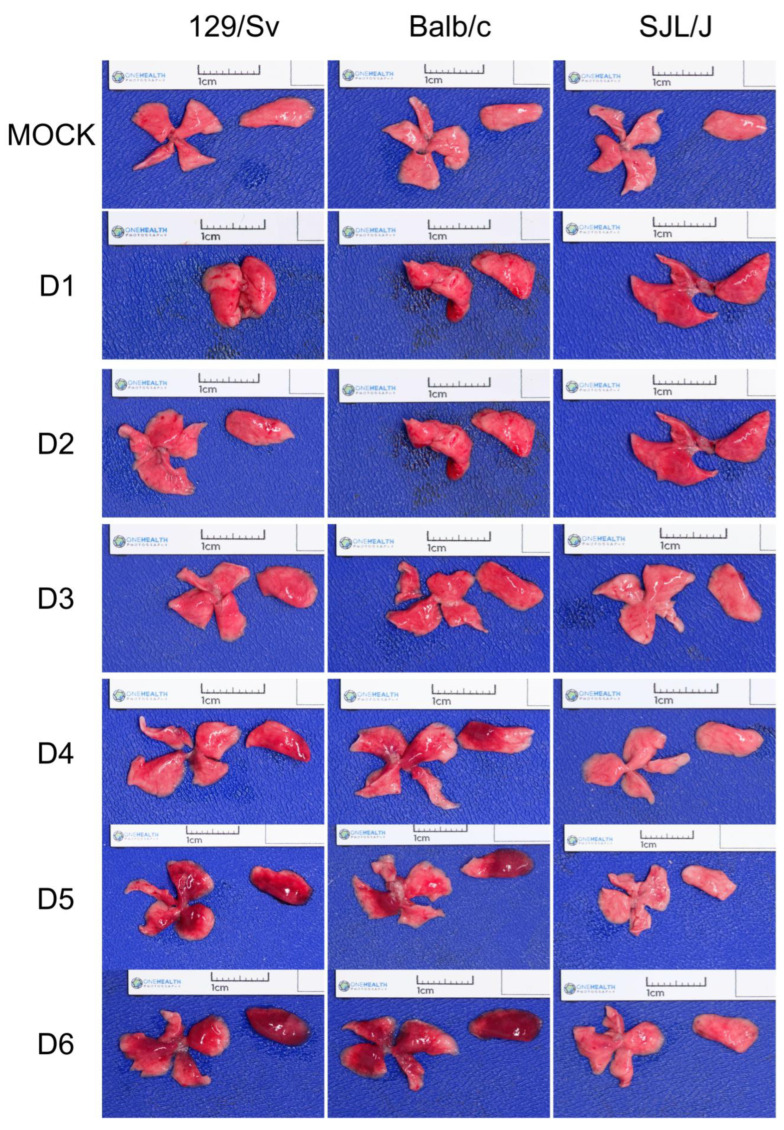
Macroscopic pictures of mice lungs. One set of lungs was chosen as a representative view among the 5 mice belonging to the same group. Both the right lung (4 lobes) and the left lung (1 lobe) are shown.

**Figure 3 viruses-17-00548-f003:**
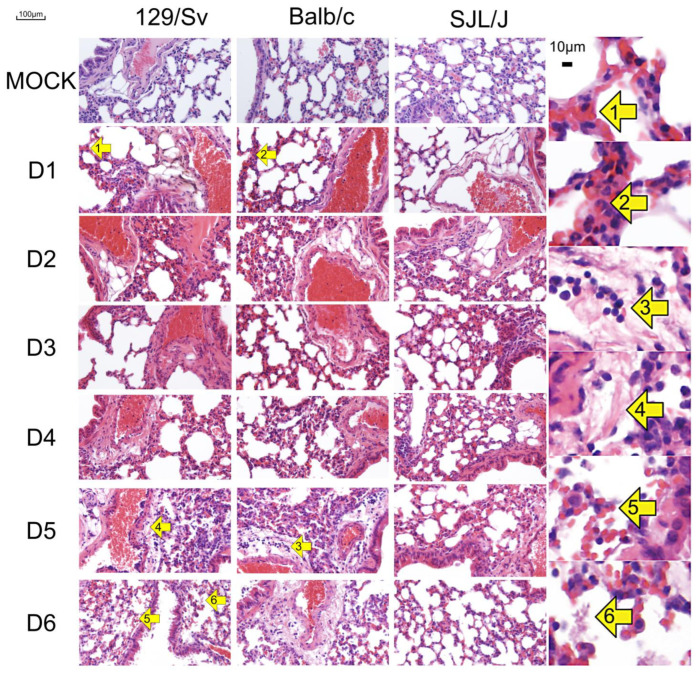
Microscopic images of mouse lungs. A representative microscopic field was selected for each mouse line from day 1 (D1) to day 6 (D6) post-infection (d.p.i.). Yellow arrows indicate (1) congestion, (2) polymorphonuclear neutrophil, (3) lymphocytes, (4) edema, (5) microhemorrhage, and (6) necrosis.

**Figure 4 viruses-17-00548-f004:**
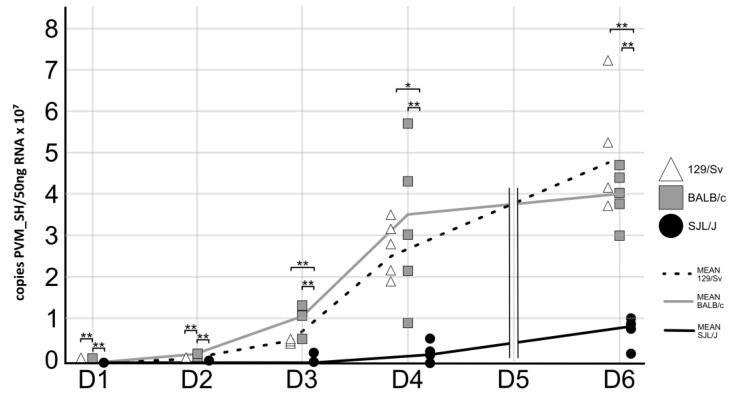
Absolute quantification of PVM particles in lung homogenates from 129/Sv (white triangles), BALB/c (gray squares), and SJL/J (black circles) during the course of the infection (*n* = 5 mice/mouse strain/time point). Statistical significance of differences between lines was calculated at each time point by one-way ANOVA with *post hoc* Tukey HSD and is indicated above the corresponding groups (* = *p* ≤ 0.05, ** = *p* ≤ 0.01).

**Figure 5 viruses-17-00548-f005:**
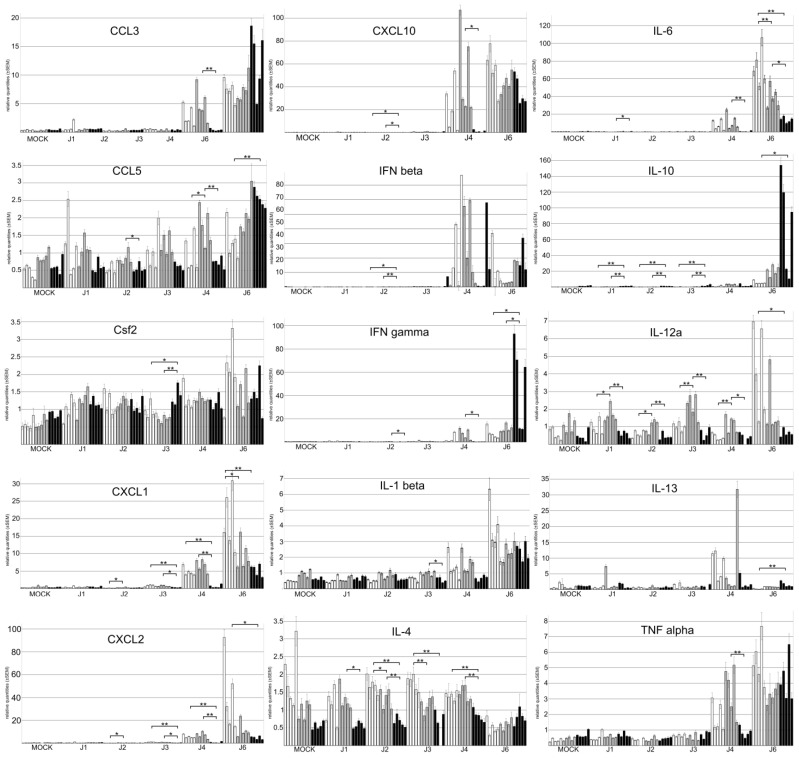
Relative quantification of 15 cytokines by qRTPCR on 1–4 and 6 dpi. Each bar = one individual. White bars = 129/Sv mice, gray bars = BALB/c mice, and black bars = SJL/J mice. Error bars = standard error of the mean. The statistical significance of differences between lines was calculated for each cytokine at each time point and is indicated above the corresponding histogram bars (* = *p* ≤ 0.05, **= *p* ≤ 0.01).

**Figure 6 viruses-17-00548-f006:**
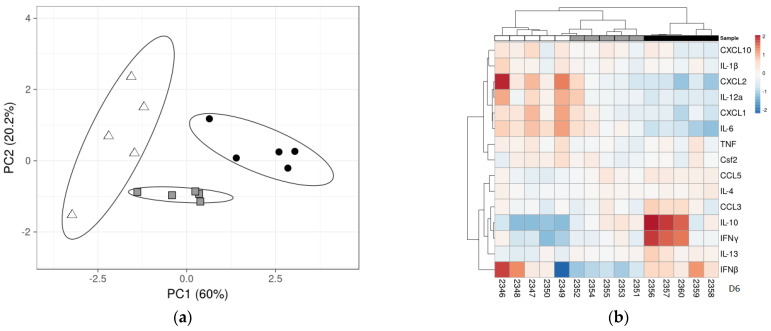
(**a**) PCA distinguishes mouse lines from each other. PCA scatterplots at 6 d.p.i.: each dot represents 1 mouse, shape- and color-coded by inbred line. White triangles = 129/Sv, gray squares = BALB/c, and black circles = SJL/J. Prediction ellipses are such that with a probability of 0.85, a new observation from the same group will fall inside the ellipse. *N* = 15 data points. (**b**) Heatmaps showing the levels of cytokines expressed in lungs of PVM-infected mice 6 d.p.i. The mean values of each cytokine were calculated and compared between mice of different inbreed origins (white = 129/Sv, gray = BALB/c, and black = SJL/J). Colors (red: upregulated, blue: downregulated) represent the deviation of the mean expression for each gene, independent of the line. All mice but one are separated according to their line, as indicated by the unsupervised clustering above the heatmap.

**Table 1 viruses-17-00548-t001:** Determination of the clinical score according to the evaluated parameters. The cutoff point was set to 6 for ethical reasons.

Parameters	Descriptions	Score
Appearance	Bright eyes, shiny and maintained coat	0
	Shaggy, dull coat	1
	Hunched, pilo-erection, disheveled (nape)	2
Hands-off monitoring	Mouse is active, interacts with its environment	0
	Slight decrease in activity, less interaction	1
	Pronounced decrease in activity, mouse is isolated	2
	Labored breathing	2
	Tremors	2
	Hyperactivity or immobility	3
	Discordant respiratory pattern	6
	Self-harm	6
Response to external stimuli	Mouse moves away quickly	0
	Mouse moves slowly or reacts excessively	1
	Delayed movement or reaction	2
	No movement, no reaction	6
Weight	Normal	0
	Loss > 10% of base weight	1
	Loss > 20% of base weight	6
Temperature	<36 °C	3
	<32 °C	6

**Table 2 viruses-17-00548-t002:** Primer and probe sequences used for cytokine quantification by qRT-PCR.

Gene	Reference	Primer1	Primer2	Probe
*Ccl3*	Mm.PT.58.29283216	CGATGAATTGGCGTGGAATC	CCTTGCTGTTCTTCTCTGTACC	/5SUN/ACTGCCTGC/ZEN/TGCTTCTCCTACAG/3IABkFQ/
*Ccl5*	Mm.PT.58.43548565	CCTCTATCCTAGCTCATCTCCA	GCTCCAATCTTGCAGTCGT	/5SUN/TCTTCTCTG/ZEN/GGTTGGCACACACTT/3IABkFQ/
*Csf2*	Mm.PT.58.9186111	CCTTGAGTTTGGTGAAATTGCC	GTCTCTAACGAGTTCTCCTTCA	/56-FAM/CGAATATCT/Zen/TCAGGCGGGTCTGCA/3IABkFQ/
*Cxcl1*	Mm.PT.58.42076891	GTGCCATCAGAGCAGTCT	CCAAACCGAAGTCATAGCCA	/56-FAM/AGGTGTCCC/Zen/CAAGTAACGGAGAAAGA/3IABkFQ/
*Cxcl10*	Mm.PT.58.43575827	TGATTTCAAGCTTCCCTATGGC	ATTTTCTGCCTCATCCTGCT	/56-FAM/ATCCCTCTC/Zen/GCAAGGACGGTC/3IABkFQ/
*Cxcl2*	Mm.PT.58.10456839	CTTTCCAGGTCAGTTAGCCTT	CAGAAGTCATAGCCACTCTCAAG	/56-FAM/CCCCCTGGT/Zen/TCAGAAAATCATCCAAAAG/3IABkFQ/
*Gapdh*	Mm.PT.39a.1	GTGGAGTCATACTGGAACATGTAG	AATGGTGAAGGTCGGTGTG	/56-FAM/TGCAAATGG/Zen/CAGCCCTGGTG/3IABkFQ/
*IFNβ*	designed in house	AGTTACACTGCCTTTGCCATC	GTCTGCTGGTGGAGTTCATC	/56-FAM/ACAATTTCT/ZEN/CCAGCACTGGGTGGA/3IABkFQ/
*IFNγ*	Mm.PT.58.41769240	TCCACATCTATGCCACTTGAG	CTGAGACAATGAACGCTACACA	/56-FAM/TCTTGGCTT/Zen/TGCAGCTCTTCCTCA/3IABkFQ/
*IL-4*	Mm.PT.58.32703659	TCTTTAGGCTTTCCAGGAAGTC	GAGCTGCAGAGACTCTTTCG	/56-FAM/AGCTGCACC/ZEN/ATGAATGAGTCCAAGT/3IABkFQ/
*IL-10*	Mm.PT.58.13531087	ATGGCCTTGTAGACACCTTG	GTCATCGATTTCTCCCCTGTG	/56-FAM/ATCACTCTT/Zen/CACCTGCTCCACTGC/3IABkFQ/
*IL-12a*	Mm.PT.58.13818295	CTCTCGTTCTTGTGTAGTTCCA	ACAGATGACATGGTGAAGACG	/5SUN/TGGTTTGGT/ZEN/CCCGTGTGATGTCTT/3IABkFQ/
*IL-1β*	Mm.PT.58.41616450	CTCTTGTTGATGTGCTGCTG	GACCTGTTCTTTGAAGTTGACG	/5SUN/TTCCAAACC/ZEN/TTTGACCTGGGCTGT/3IABkFQ/
*IL-6*	Mm.PT.58.10005566	TCCTTAGCCACTCCTTCTGT	AGCCAGAGTCCTTCAGAGA	/56-FAM/CCTACCCCA/Zen/ATTTCCAATGCTCTCCT/3IABkFQ/
*IL-13*	Mm.PT.58.11338747	GAATCCAGGGCTACACAGAAC	AACATCACACAAGACCAGACTC	/56-FAM/TCCACACTC/ZEN/CATACCATGCTGCC/3IABkFQ/
*Rplp0*	Mm.PT.58.43894205	CGCTTGTACCCATTGATGATG	TTATAACCCTGAAGTGCTCGAC	/56-FAM/AGGCCCTGC/Zen/ACTCTCGCTT/3IABkFQ/
*Tnf*	Mm.PT.58.12575861	TCTTTGAGATCCATGCCGTTG	AGACCCTCACACTCAGATCA	/5SUN/CCACGTCGT/ZEN/AGCAAACCACCAAGT/3IABkFQ/

**Table 3 viruses-17-00548-t003:** Scoring of microscopic lesions observed during the progression of the disease. White columns represent 129/Sv mice, gray columns represent BALB/c mice, and black columns represent SJL/J mice. Scoring goes from 0: no lesion to 3: severe lesion. The values correspond to the means for each group of mice (*n* = 5), along with the standard deviation (SD).

Lesion		Mock			D1			D2			D3			D4			D5			D6		

Congestion		0.5 ±0	0.5 ±0	0.5 ±0	0.8 ±0.6	0.5 ±0.4	0.5 ±0.3	1.7 ±0.7	1.1 ±0.6	0.6 ±0.2	1.4 ±0.7	1.3 ±0.4	0.9 ±0.3	2.4 ±0.5	1.8 ±0.4	2 ±0	2.1 ±0.3	1.4 ±0.9	1.5 ±0.4	2.1 ±0.6	1.7 ±0.6	1.3 ±0.9
Interstitial inflammation		0 ±0	0 ±0	0 ±0	0.1 ±0.2	0.2 ±0.2	0 ±0	0.8 ±0.2	0.6 ±0.2	0.6 ±0.2	1.1 ±0.4	1.2 ±0.5	0.7 ±0.2	1.6 ±0.5	2 ±0.6	1 ±0	1.3 ±0.2	1 ±0.5	0.5 ±0	1.9 ±0.4	1.9 ±0.5	1 ±0
Perivascular inflammation		0 ±0	0 ±0	0 ±0	0 ±0	0 ±0	0 ±0	0.1 ±0.2	0.2 ±0.2	0.2 ±0.4	0.4 ±0.3	0.6 ±0.3	0.4 ±0.3	1.8 ±0.4	1.4 ±0.5	0 ±0	0.9 ±0.2	1.2 ±0.2	0.3 ±0.2	1.4 ±0.7	1.7 ±0.6	0.6 ±0.8
Perivascular edema		0 ±0	0 ±0	0 ±0	0 ±0	0 ±0	0 ±0	0.1 ±0.2	0.4 ±0.4	0.2 ±0.2	0.2 ±0.4	0.6 ±0.3	0.4 ±0.3	1 ±0.6	1.4 ±0.5	0 ±0	0.7 ±0.2	1.2 ±0.2	0.3 ±0.2	1.7 ±1	2 ±0.5	0.2 ±0.4
Parenchymal inflammation		0 ±0	0 ±0	0 ±0	0 ±0	0 ±0	0 ±0	0 ±0	0 ±0	0 ±0	0.5 ±0.4	0.2 ±0.2	0 ±0	0.4 ±0.5	1.4 ±0.8	0 ±0	1.3 ±0.2	1.9 ±0.5	0.3 ±0.2	1.5 ±0.6	1.7 ±0.6	0.4 ±0.5
Micro hemorrhages		0 ±0	0 ±0	0 ±0	0 ±0	0 ±0	0 ±0	0 ±0	0 ±0	0 ±0	0.4 ±0.3	0.3 ±0.4	0 ±0	0.6 ±0.5	1.4 ±0.8	0 ±0	1.3 ±0.2	1.9 ±0.5	0 ±0	1.7 ±0.7	1.6 ±0.6	0.6 ±0.8
Necrosis		0 ±0	0 ±0	0 ±0	0 ±0	0 ±0	0 ±0	0 ±0	0 ±0	0 ±0	0.3 ±0.4	0.3 ±0.4	0 ±0	0.4 ±0.5	1.2 ±0.7	0 ±0	1 ±0.4	1.9 ±0.5	0 ±0	1.3 ±0.6	1.6 ±0.5	0.4 ±0.5
TOTAL	Mean	0.5	0.5	0.5	0.9	0.7	0.5	2.7	2.3	1.6	4.3	4.5	2.4	8.2	10.6	3	8.6	10.5	2.9	11.5	12.1	4.5
	SD	±0	±0	±0	±0.6	±0.7	±0.3	±1.2	±1.2	±1	±2.7	±2.4	±1.2	±1.9	±3.9	±0	±1.4	±2.7	±0.3	±2.1	±2.9	±2

**Table 4 viruses-17-00548-t004:** Cytokines with the largest loadings for PC1 and PC2 from day 1 to 6 p.i.

Day p.i.	PC1	PC2	PC1—Largest Loadings	PC2—Largest Loadings
1	36	30.7	IL-13 (−0.95)	IL-12a (−0.51)
			CCL5 (0.20)	IL-10 (0.45)
			IL-10 (−0.14)	IL-4 (−0.43)
2	52.3	18	IL-10 (0.65)	IL-12a (−0.66)
			IL-4 (−0.50)	IL-1β (−0.38)
			IFNβ (0.45)	CCL5 (−0.34)
3	43.8	18.7	IL-10 (−0.68)	IL-12a (−0.59)
			IL-12a (0.41)	CXCL10 (−0.43)
			CCL5 (0.26)	IL-10 (−0.41)
4	78.4	13.1	CXCL10 (0.55)	IL-13 (−0.71)
			IFNβ (0.50)	IFNβ (0.43)
			IL-6 (0.35)	CXCL2 (−0.25)
6	60	20.2	IL-10 (−0.54)	IFNβ (−0.94)
			IFNγ (−0.42)	CXCL2 (−0.21)
			CXCL2 (0.41)	IL-10 (0.18)

**Table 5 viruses-17-00548-t005:** Pooled correlation analyses between microscopic lesion score (Micro.), pulmonary viral load (V. load), and 15 cytokines. The values in the lower left half of the table correspond to Pearson’s r and the values in the upper right half of the table correspond to the associated *p*-values (after Bonferroni correction). Pearson’s *r* values > 0.8 are highlighted in bold. Analyses were performed on pooled data from 129/Sv, BALB/c, and SJL/J mice (*n* = 89).

	Micro.	V. load	IFNβ	CCL3	CCL5	CXCL1	CXCL10	CXCL2	Csf2	IL-6	IL-1β	TNFα	IFNγ	IL-10	IL-12a	IL-13	IL-4
Micro.		1.3 × 10^−18^	6.8 × 10^−3^	5.5 × 10^−5^	1.4 × 10^−2^	6.6 × 10^−12^	3.4 × 10^−12^	1.9 × 10^−5^	2.1 × 10^−1^	2.0 × 10^−10^	4.4 × 10^−9^	2.9 × 10^−10^	1	1	9.1 × 10^−3^	1	1
V. load	**0.839**		4.0 × 10^−6^	7.4 × 10^−5^	7.2 × 10^−3^	6.4 × 10^−15^	2.6 × 10^−18^	3.1 × 10^−5^	1.5 × 10^−2^	5.0 × 10^−14^	3.6 × 10^−8^	8.2 × 10^−14^	1	1	1.4 × 10^−2^	1	1
IFNβ	0.459	0.598		4.7 × 10^−4^	5.4 × 10^−4^	8.7 × 10^−2^	1.5 × 10^−12^	2.4 × 10^−2^	1	6.0 × 10^−1^	2.6 × 10^−3^	7.1 × 10^−8^	1	1	1	1	1
CCL3	0.549	0.544	0.516		2.2 × 10^−13^	1.8 × 10^−6^	4.6 × 10^−15^	8.5 × 10^−3^	8.7 × 10^−1^	5.3 × 10^−6^	1.3 × 10^−8^	2.9 × 10^−16^	7.5 × 10^−20^	1.3 × 10^−15^	1	1	5.3 × 10^−1^
CCL5	0.435	0.452	0.514	0.767		1.1 × 10^−1^	1.4 × 10^−8^	1	1	1.7 × 10^−1^	6.0 × 10^−4^	9.0 × 10^−8^	8.2 × 10^−7^	2.4 × 10^−5^	1	1	1
CXCL1	0.74	0.791	0.393	0.603	0.382		8.8 × 10^−14^	2.3 × 10^−14^	5.4 × 10^−9^	1.0 × 10^−38^	4.4 × 10^−12^	1.7 × 10^−20^	1	1	4.7 × 10^−10^	1	2.8 × 10^−1^
CXCL10	0.746	**0.836**	0.759	0.794	0.665	0.774		6.5 × 10^−6^	5.1 × 10^−2^	5.9 × 10^−12^	5.8 × 10^−10^	6.8 × 10^−24^	4.4 × 10^−3^	1.7 × 10^−1^	3.0 × 10^−2^	1	1
CXCL2	0.567	0.559	0.428	0.448	0.295	0.783	0.584		3.6 × 10^−1^	4.8 × 10^−16^	2.5 × 10^−13^	7.4 × 10^−8^	1	1	7.6 × 10^−17^	1	1
Csf2	0.361	0.435	0.148	0.314	0.105	0.676	0.403	0.345		9.4 × 10^−7^	1.1 × 10^−1^	1.7 × 10^−6^	1	1	1	1	1
IL-6	0.71	0.778	0.332	0.587	0.368	**0.958**	0.741	**0.808**	0.612		1.0 × 10^−10^	6.5 × 10^−16^	1	1	2.4 × 10^−11^	1	1.8 × 10^−2^
IL-1β	0.678	0.654	0.481	0.666	0.505	0.744	0.699	0.766	0.381	0.716		3.5 × 10^−12^	1.7 × 10^−1^	1	5.9 × 10^−5^	1	1
TNFα	0.709	0.777	0.656	**0.814**	0.646	**0.862**	**0.891**	0.649	0.607	**0.809**	0.749		5.6 × 10^−3^	1.3 × 10^−1^	1.9 × 10^−3^	1	1
IFNγ	0.201	0.146	0.288	**0.853**	0.614	0.232	0.463	0.148	0.045	0.222	0.369	0.461		7.9 × 10^−56^	1	1	1
IL-10	0.147	0.068	0.155	**0.802**	0.563	0.167	0.369	0.066	0.032	0.165	0.299	0.38	**0.986**		1	1	1
IL-12a	0.449	0.439	0.192	0.255	0.252	0.705	0.42	**0.821**	0.295	0.733	0.551	0.487	0.01	−0.053		1	1
IL-13	0.197	0.108	0.054	0.005	0.002	0.019	0.071	−0.038	0.024	−0.069	0.156	0.014	−0.012	−0.014	−0.140		1
IL-4	−0.141	−0.204	0.027	−0.334	−0.128	−0.355	−0.247	−0.259	−0.265	−0.433	−0.237	−0.313	−0.238	−0.254	−0.186	0.181	

## Data Availability

The data presented in this study are included in the article. Further inquiries can be directed to the corresponding author.

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
