# Peer review of "Host-Strain-Specific Responses to Pneumonia Virus of Mice Infection: A Study of Lesions, Viral Load, and Cytokine Expression"

_viruses, 2025, doi:10.3390/v17040548_

Round 1

Reviewer 1 Report

Comments and Suggestions for Authors

This manuscript describes a study in which macroscopic lesions, histopathology, viral load and cytokine expression were analysed in the lungs of mice with different genetic backgrounds, from 1 to 6 days after infection with pneumonia virus of mice (PVM). The 3 mouse lines used were those that had been shown in previous studies to vary in their susceptibility to PVM infection, i.e. SJL/J mice were resistant, BALB/c mice had intermediate susceptibility and 129/Sv mice were susceptible. The results indicated a strong correlation between viral load and microscopic lung lesions in 129/Sv, BALB/c, and SJL/J mouse lines, and a significant correlation between cytokine levels and the severity of microscopic lung lesions in 129/Sv and BALB/c mice.  The authors concluded that the pathogenicity of PVM is partially driven by the virus itself and that there is an important role for cytokines in disease progression.  However, there did not appear to be any difference in the susceptibility of 129/Sv and BALB/c mice to PVM infection as clinical scores, macroscopic and microscopic lung lesions & viral load appeared to be very similar in these 2 mouse strains. Despite this apparent similar susceptibility to PVM infection, the day 6 cytokine scatter plot and heat map in Fig. 6 revealed differences in the cytokine responses in 129/Sv and BALB/c lines of mice. This does not appear to have been adequately discussed. Similarly, the possible mechanisms responsible for the low levels of virus replication and little lung pathology in SJL/L mice despite the “heightened levels of TNF-α and IFNγ production, resulting in enhanced expression of CXCL10 and CCL3” at day 6 p.i. have not been adequately discussed.

The findings reported in the manuscript should be discussed in the context of other studies that have analysed the cytokine responses in PVM-infected mice. For example, the cytokine profile in the lungs of PVM-infected BALB/c mice reported in this manuscript could be compared with that reported by Limkar et al 2021 (doi: 10.3390/v13050728) and discussed in relation to studies in PVM-infected IFNγ gene-deleted BALB/c mice, which showed that although there were no differences in virus replication or survival compared to wild-type mice, lung pathology was more severe (Glineur et al.,10.1016/j.virol.2014.07.039).

  1. Line 49: Reference [13] has been provided to show that “susceptibility to disease in general and to PVM in particular, differs dramatically among inbred mouse strains”. However, this reference describes a genotyping method for mice and is not about susceptibility of mice to PVM.
  2. Paragraph 2.5: This should include some information about the scoring used for analysis of microscopic lung lesions. The only information about the scoring is in the Legend to Table 2, which states “Scoring goes from 0: no lesion to 3: severe lesion. More details are required.
  3. Paragraph 2.6 qRT-PCR: This should include some information about the primers and probes used to analyse cytokine expression.
  4. Table 2 should include the standard deviation of the mean values of the microscopic lesions.
  5. 3, 10 µm images: These images are not in focus making it difficult to see the high-power histopathology.
  6. 4 indicates that there are statistically significant differences in lung virus load between 129/v and BALB/c mice and between BALB/c and SJL/J mice at day 1 and 2 post-infection (p.i.). However, the differences in viral load at these times is not obvious from the figure. The authors should consider presenting the means ± sd of the values at these times p.i. in the text.
  7. 5: The data presented in this figure is difficult to interpret. In many of the figures it is not possible to see individual white histograms. It would be much clearer if the data is presented as the mean ± standard deviation of each cytokine for each mouse strain at each time point. It is especially difficult to appreciate the significance of differences in cytokine expression levels between mouse strains at early time points, when the levels are low but are highlighted to show statistically significant differences. Are these differences likely to have biological significance? The data could be shown using a broken y-axis as shown in figs. 1A, 1B & 1E of the paper cited as reference [20].
  8. Lines 320-322: This states “Cytokine storms have also been shown to determine the clinical severity and consequently the lethality of other respiratory diseases i.e. SARS, MERS, H1N1 influenza A, Spanish flu [19], and hRSV [20].” However, reference [20] refers to a study of PVM in mice and not hRSV.
Comments on the Quality of English Language

The English could be improved in a few places e.g. "weighted" (line 95) should be "weighed"; "facies" (line 100) should presumably ne "faces"; "euthanasy" (line 177) should be "euthanasia"; it is not clear what "sensible" (line 201) means.

PRC (line 137) should be "PCR"

Author Response

Dear Reviewer,

We sincerely appreciate the time and effort you have dedicated to reviewing our manuscript. Your thoughtful and constructive comments have been invaluable in improving the clarity and quality of our work. We are especially grateful for your insightful suggestions, which have allowed us to refine our manuscript further.

In response to your recommendations, we have revised the discussion to better contextualize our findings within previous studies that have analyzed cytokine responses in PVM-infected mice.

Regarding the scatter plot at day 6 p.i., we have clarified and emphasized the significant differences in susceptibility between 129/Sv and BALB/c mice at various levels:

  • Clinical score at day 6 p.i.: We have expanded on this in lines 190–192:
    "These differences were more pronounced on day 6 p.i., with fourteen 129/Sv, four BALB/c, but no SJL/J mice reaching a cumulative clinical score exceeding six—i.e., the threshold for euthanasia (Fig. 1)."
    Additionally, we have explicitly stated that the differences in clinical scores between 129/Sv and BALB/c mice are statistically significant (One-way ANOVA with post-hoc Tukey HSD test), a point that was previously omitted. This clarification has been added in lines 192–195.
  • Viral load at days 1 and 2 p.i.: We have revised lines 241–245 to include precise viral load values for the different mouse strains, as per your suggestion in comment #6:
    "Strikingly, significant differences between strains were already evident at 1 d.p.i. At this time point, BALB/c mice exhibited significantly higher viral loads in the lungs compared to 129/Sv and SJL/J mice (14,850 ± 4,403; 3,299 ± 1,573; and 7,552 ± 2,413, respectively). This difference persisted at 2 d.p.i., with viral loads of 1,197,665 ± 275,091 in BALB/c, 428,088 ± 93,922 in 129/Sv, and 190,067 ± 51,039 in SJL/J."
    This revision emphasizes the differences between BALB/c and 129/Sv mice, improving clarity and completeness.

Please find our detailed, point-by-point responses to your comments below. We hope these revisions address your concerns and strengthen the manuscript.

Once again, we sincerely appreciate your time, expertise, and valuable feedback.

  1. Line 49: Reference [13] has been provided to show that “susceptibility to disease in general and to PVM in particular, differs dramatically among inbred mouse strains”. However, this reference describes a genotyping method for mice and is not about susceptibility of mice to PVM.

Thank you for noticing this mistake. We were planning to cite a publication discussing different susceptibility to viral infection between 2 inbred mouse lines used as animal model. We have made the changes in line 49 and in the reference section.

  1. Paragraph 2.5: This should include some information about the scoring used for analysis of microscopic lung lesions. The only information about the scoring is in the Legend to Table 2, which states “Scoring goes from 0: no lesion to 3: severe lesion. More details are required.

Etienne

  1. Paragraph 2.6 qRT-PCR: This should include some information about the primers and probes used to analyse cytokine expression.

A table with the sequences of the primers and probes used to analyse cytokine expression has been added in the Material and Method section (Table 2). The numbering of the other tables has been changed accordingly.

  1. Table 2 should include the standard deviation of the mean values of the microscopic lesions.

We have added this information.

  1. 3, 10 µm images: These images are not in focus making it difficult to see the high-power histopathology.

This limitation arises from the resolution of our microscope. To address this issue, we enhanced the sharpness of the images using DxO PhotoLab and Serif Affinity Photo 2. This information has been added to the Materials and Methods section and the new, sharper, version of the figure has been inserted in the manuscript.

  1. 4 indicates that there are statistically significant differences in lung virus load between 129/v and BALB/c mice and between BALB/c and SJL/J mice at day 1 and 2 post-infection (p.i.). However, the differences in viral load at these times is not obvious from the figure. The authors should consider presenting the means ± sd of the values at these times p.i. in the text.

Thank you for this suggestion, we have added the values in the text.

  1. 5: The data presented in this figure is difficult to interpret. In many of the figures it is not possible to see individual white histograms. It would be much clearer if the data is presented as the mean ± standard deviation of each cytokine for each mouse strain at each time point. It is especially difficult to appreciate the significance of differences in cytokine expression levels between mouse strains at early time points, when the levels are low but are highlighted to show statistically significant differences. Are these differences likely to have biological significance? The data could be shown using a broken y-axis as shown in figs. 1A, 1B & 1E of the paper cited as reference [20].

The white histograms were indeed difficult to distinguish in the manuscript file although they were visible in the original figure when viewed separately. We suspect that the resolution decreased when the figure was embedded in the combined file. We have now solved this issue.

Regarding the data presentation, we have chosen to display individual values for each mouse rather than presenting the mean ± standard deviation to preserve information on interindividual variability. While we acknowledge that this format makes the data more challenging to interpret at a glance, we believe it provides a more accurate representation of biological variability.

To improve readability, particularly at early time points where cytokine levels are low but statistically significant differences are highlighted, we have generated an additional figure using a logarithmic scale. We propose including this figure in the supplemental information (supplemental figure 2). We opted for this approach instead of a broken y-axis because, for some cytokines, breaking the axis would have resulted in the loss of upper portions of histograms for certain mice.

Additionally, upon further reflection, we recognized that ANOVA is a more appropriate statistical method for analyzing cytokine expression differences between mouse lines than pairwise t-tests. ANOVA allows for a global comparison across groups while accounting for overall variability, thereby reducing the risk of Type I errors associated with multiple comparisons. Moreover, this approach enables a more precise identification of significant differences through appropriate post-hoc tests.

Consequently, we have reanalyzed our data using ANOVA, followed by post-hoc Tukey HSD tests where applicable, and have updated the manuscript accordingly. We hope these adjustments enhance the clarity, robustness, and relevance of our findings.

  1. Lines 320-322: This states “Cytokine storms have also been shown to determine the clinical severity and consequently the lethality of other respiratory diseases i.e. SARS, MERS, H1N1 influenza A, Spanish flu [19], and hRSV [20].” However, reference [20] refers to a study of PVM in mice and not hRSV.

Reference 20 is indeed a study on PVM. This study states, “The cytokine storm is a major component of severe respiratory infections, such as those caused by hRSV; consequently, targeting the host immune response is an alternative strategy (6–8).” To ensure proper attribution, we have replaced reference 20 with the original source of this statement, reference 8:

Tisoncik, J. R., Korth, M. J., Simmons, C. P., Farrar, J., Martin, T. R., & Katze, M. G. (2012). Into the eye of the cytokine storm. Microbiology and Molecular Biology Reviews, 76(1), 16–32. https://doi.org/10.1128/MMBR.05015-11.

.

Comments on the Quality of English Language

The English could be improved in a few places e.g. "weighted" (line 95) should be "weighed" (corrected); "facies" (line 100) should presumably ne "faces" (changed to facial appearance); "euthanasy" (line 177) should be "euthanasia"(corrected); it is not clear what "sensible" (line 201) means (changed to “susceptible”).

PRC (line 137) should be "PCR"

This has been corrected.

Reviewer 2 Report

Comments and Suggestions for Authors

Pneumonia virus of mice (PVM) is a paramyxovirus that is regarded as the animal model for human infection by respiratory syncytial virus (hRSV), as it induces many of the severe clinical and pathologic symptoms that characterize hRSV infection in infants.  One understudied aspect that may be determinative with respect to hRSV disease severity is the influence of patient genetic background.  In this study, the goal is to exploit the observed dramatic differences in susceptibility to PMV among inbred mouse strains to begin to understand the basis for the genetic background-related different RSV disease patterns in humans.

To address this issue, the authors compare the clinical outcomes both macroscopically and microscopically in the lungs of three different strains, including the resistant SJL/J, intermediate BALB/c and susceptible 129/Sv strains.  This followed by a comparison of the expression profiles of 15 cytokines in the lungs of infected animals of each strain.  The goal is to try to establish a correlation between differences in the expression patterns of specific cytokines and disease severity.

The primary findings of the study are that: 1) disease severity is determined to a significant extent by the genetic background of the infected animal; 2) a strong correlation can also be demonstrated between viral load and microscopic lesions in the three different strains, indicating that PVM pathogenicity is partially determined by the virus itself; and, 3) a significant correlation exists between the levels of specific cytokines (CCL3, CXCL1, CXCL10, TNF and IFN-g) and disease severity.  Based on these findings, the authors conclude that viral load and cytokine-related damage, as well as genetic background, influence disease severity to different extents. 

The authors duly note that their findings with respect to the impact of the “cytokine storm” in PVM disease severity is not that novel, having been established for multiple viruses.  The same certainly applies to the demonstration of a link between viral load and disease severity.  However, although there is evidence that genetic background of patients may influence the severity of RSV disease, this study contributes a definitive demonstration of the concerted influence of genetic background, viral load and cytokine levels on disease severity.  Given the recognized suitability of PVM as an animal model for hRSV, these findings strongly suggest that the same interplay may be operative in the latter disease.

This is considered an exceptionally strong study that provides a definitive demonstration of the interplay between genetic background, viral load and cytokine levels in the severity of pneumovirus-induced disease in an animal model.  The authors do an outstanding job of presenting an extensive set of data in a direct and well-organized format.  The data are all carefully and appropriately evaluated and support the conclusions made. 

Author Response

Dear Reviewer,

We sincerely appreciate the time and effort you dedicated to reviewing our manuscript. Your thoughtful feedback and very positive comments mean a great deal to us. We are grateful for your support and for recognizing the value of our work.

Thank you again for your time and expertise.

Reviewer 3 Report

Comments and Suggestions for Authors

The present study ("Host Strain-Specific Responses to Pneumonia Virus of Mice Infection: A Study of Lesions, Viral Load, and Cytokine Expression") by Levy et al. is quite extensive, the manuscript is well-written, and the results are quite promising as well. I, therefore, recommend it for publication in the journal Viruses.

Author Response

(The authors gave the same response as above.)

Round 2

Reviewer 1 Report

Comments and Suggestions for Authors

The authors have addressed the majority of the comments raised and discussed their findings in a wider context. Although there appears to be a significant difference in clinical score between 129/Sw (reportedly susceptible) and BALB/c mice (reportedly intermediate susceptibility), 6 days post-infection, the extent of macroscopic and microscopic lung lesions & viral load at this time point appeared to be very similar in these 2 mouse strains.  Despite this apparent similarity in microscopic lesion scores & viral load in 129/Sw and BALB/c mice 6 days after PVM infection, the day 6 cytokine scatter plot and heat map in Fig. 6 revealed differences in the cytokine responses in these two mouse lines. This still has not been adequately discussed.

The supplementary Fig. 2 showing cytokine data on a log scale more clearly highlights differences in cytokine expression at early time-points. However, it is difficult to see the white histograms.  Is it possible to improve the resolution of this figure?

Author Response

Dear Reviewer,

We would like to express our sincere thanks once again for the time and effort you dedicated to reviewing our manuscript.

In response to the question you raised regarding the scatter plot at 6 d.p.i., we have expanded the discussion to address this point (see lines 398-404).

Regarding the visibility of the white histograms in Figure 5 and Supplemental Figure 2, we have sent the original, non-embedded files to our assistant editor, Ms. Thea Yuan. She will forward them to you and coordinate with the person responsible for printing to ensure that the figures meet the required quality standards.

Thank you again for your valuable feedback.

Yours sincerely,

Anne-Sophie Van Laere